# Phylogenic classification and virulence genes profiles of uropathogenic *E. coli* and diarrhegenic *E. coli* strains isolated from community acquired infections

Rasha M. Khairy[1]*, Ebtisam S. Mohamed[1], Hend M. Abdel Ghany[2], Soha S. Abdelrahim[1,3]

1 Department of Microbiology and Immunology, Faculty of Medicine, Minia University, Minia, Egypt, 2 Department of Biochemistry, Faculty of Medicine, Minia University, Minia, Egypt, 3 Department of Biomedical Sciences, College of Medicine, King Faisal University, Al-Ahsa, Saudi Arabia

* rashakhiry1@gmail.com

**Data Availability Statement:** All relevant data are within the paper and its Supporting Information files.

## Abstract

The emergence of *E.coli* strains displaying patterns of virulence genes from different pathotypes shows that the current classification of *E.coli* pathotypes may be not enough, the study aimed to compare the phylogenetic groups and urovirulence genes of uropathogenic *Escherichia coli* (UPEC) and diarrheagenic *E.coli* (DEC) strains to extend the knowledge of *E.coli* classification into different pathotypes. A total of 173 UPEC and DEC strains were examined for phylogenetic typing and urovirulence genes by PCR amplifications. In contrast to most reports, phylogenetic group A was the most prevalent in both UPEC and DEC strains, followed by B2 group. Amplification assays revealed that 89.32% and 94.29% of UPEC and DEC strains, respectively, carried at least one of the urovirulence genes, 49.5% and 31.4% of UPEC and DEC strains, respectively, carried ≥ 2 of the urovirulence genes, *fim H* gene was the most prevalent (66.9% and 91.4%) in UPEC and DEC strains respectively. Twenty different patterns of virulence genes were identified in UPEC while 5 different patterns in DEC strains. Strains with combined virulence patterns of four or five genes were belonged to phylogenetic group B2. Our finding showed a closer relationship between the DEC and UPEC, so raised the suggestion that some DEC strains might be potential uropathogens. These findings also provide different insights into the phylogenetic classification of *E. coli* as pathogenic or commensals where group A can be an important pathogenic type as well as into the classification as intestinal or extra- intestinal virulence factors.

## Introduction

*Escherichia coli* are normal inhabitants of gastrointestinal tract of humans and many animals, however some *E.coli* strains acquired specific virulence genes that enable them to cause intestinal and extra-intestinal infections in humans such as diarrhea and urinary tract infections [1]. Diarrheagenic *E.coli* (DEC) represent a leading bacterial cause of diarrhea all over the world.

**Funding:** The authors received no specific funding for this work.

**Competing interests:** The authors have declared that no competing interests exist.

DEC pathotypes are characterized by their specific virulence determinants [2]. *E.coli* strains causing extra-intestinal infections are known as extra-intestinal pathogenic *E.coli* (ExPEC) [3]. Molecular and epidemiological studies have identified ExPEC as a distinct *E. coli* type. ExPEC strains usually carry characteristic virulence factors that allow colonization of the host mucosa and conferring their pathogenic potential [4].

Urinary tract infection (UTI) is the most common extra-intestinal infection caused by *E. coli* [5], which occur mainly due to the spread of Uropathogenic Escherichia coli (UPEC) strains from the intestine to the urinary tract [6]. These strains become pathogenic by acquiring new virulence properties encoded by specific genes, allowing them to colonize host mucosal surfaces and invade the normally sterile urinary tract [5]. Surface virulence factors (adhesins) are very important virulence factor of UPEC as the main attachment factor, P fimbriae is associated with pyelonephritis and is encoded by *pap* genes. [7]. Other adhesins that act as virulence factors are S fimbrial adhesin, which is coded by sfa genes and Type 1 fimbriae which is encoded by the *fim* gene cluster[8, 9, 10,11]. A part from adhesins, important virulence factors of UPEC strains are the toxins that cause an inflammatory response. The most important secreted virulence factor is a lipoprotein toxin called $\alpha$-haemolysin (HlyA) which is encoded by *hlyA* gene [12]. Regarding phylogenetic typing, *E.coli* strains are classified into four major phylogenetic groups (phylogroups) named as A, B1, B2, and D [13], moreover these phylogenetic groups are intertwined with virulence patterns [1]. Each *E.coli* type has characteristic patterns, which allow them to colonize and invade their host [14]. However, exceptions exist where some UPEC strains have been reported to carry DEC markers [15], on the other hand, some DEC strains carry virulence factors associated with UPEC [5, 15]. Interestingly, *E.coli* strains that carry genetic determinants from different *E.coli* pathotypes are now termed as "heteropathogenic *E.coli*" [14, 16, 17], so the current classification of *E.coli* pathotypes may be not enough [18, 19]. These findings raised the suggestion that some DEC strains might be potential uropathogens. The current study has investigated a collection of DEC and UPEC strains isolated from clinical cases regarding the presence of urovirulence determinants and phylogenetic grouping. The aim was to detect whether DEC strains share virulence properties with the UPEC pathotypes and to recognize their phylogenetic diversity.

## Materials and methods

A total of 173 *E.coli* strains were included in the study were recovered from patients of both sex and different ages presenting symptomatic UTIs and diarrhea. A total of 103 isolates were recovered from urine samples of patients with UTIs (UPEC) during outpatient treatment (patients who visited an acute day ward were considered to be outpatients). *E.coli* UTI was diagnosed by clinical symptoms such frequency, urgency, dysuria, small-volume voids or lower abdominal pain in addition to urine culture with a colony count $>10^5$ CFU *E.coli*/ml in midstream urine sample. Seventy strains were recovered from the stool samples of patients with diarrhea (DEC) during outpatient treatment. Diarrhea was diagnosed by, passage of loose stools for three times or more daily, in addition to one or more of characteristic clinical symptoms (nausea, vomiting, abdominal pain or cramps, fecal urgency, or dysentery). The samples of the study were collected from outpatients' clinics, Minia University Hospitals, Egypt during the period from January to March 2018.

### Ethics statement

The study protocol was approved by Minia Faculty of Medicine Review Board (code: 46 A at 2/1/2018). Written informed consents were obtained from all patients for the use of their samples.

## Bacterial isolates

Urine samples were cultured on chromogenic media (CHROMagar™ Orientation, Paris, France), while diarrhea samples were cultured on MacConkey and EMB agar. The isolates were confirmed as *E.coli* by standard bacteriological and biochemical tests including indole, urease, citrate and sugar fermentation tests. *E. coli* as the sole urine and stool Cultures micro-organism were only included. Strains confirmed as *E. coli* were kept in trypticase soy broth with sterilized 15% glycerol at—20˚C. The DEC strains were identified as enteroaggregative *E. coli* (EAEC) by PCR technique [20].

## DNA extraction

DNA was extracted by using the QIAamp DNA extraction Mini kit (Qiagen, Hilden, Germany) according to manufacturer's instructions. DNA was used immediately or stored at—20˚C until used.

## Phylogenetic analysis

*E. coli* isolates were classified into phylogenetic groups by targeting two marker genes (*chuA* and *yjaA*) and a DNA fragment TSPE4.C2 (Table 1) by triplex PCR as described previously [21]. Additional sub-grouping scheme proposed by Branger *et al*, 2005 was used [22].

## Detection of virulence genes

Specific primers were used to amplify sequences of 6 different virulence genes. Primer sequences and predicted sizes of the PCR products are shown in (Table 1). The amplification reactions were carried out using Biometra, UNO II thermal cycler (Goettingen, Germany) under the following conditions: initial denaturation at 95˚C for 5 min, followed by 35 cycles of: 30 s at 94˚C for, 30s at 63˚C, then 30 s at 72˚C, with a final extension step at 72˚C for 5 min. PCR was performed in a 25 ml reaction mixture containing1 ul of template DNA ($^*$100 ng/ ml), 12.5 ml of PCR mastermix (Maxima Hot Start Green PCR Master Mix, USA), and 1 ul (10 pmol) of each primer and 9.5 ml of nuclease free water. PCR products were resolved on 2% agarose gel and visualized under a UV transilluminator (Biometra).

## Statistical analysis

The chi -square test or the Fisher's exact test was used. $P < 0.05$ was considered statistically significant (two-tailed).

## Results

Phylogenetic grouping and virulence genes were characterized in103 UPEC isolates and 70 DEC isolates (EAEC) using PCR assay. Regarding the phylogenetic analysis, the predominant groups were $A_1$ (31.1% UPEC, 37.1% DEC) with no significant difference (p value = 0.723); $A_0$ (23.3%UPEC, 12.9%DEC; p value = 0.068) followed by group $B_2$ then group $D_1$. Phylogenetic groups $D_2$ and $B_1$ were detected only in UPEC isolates and were not detected in DEC isolates. There were no significant difference between phylogenetic distribution in UPE and DEC isolates (Fig 1 and S1 Fig). Six virulence genes were examined in103 UPEC isolates and 70 DEC isolates (EAEC) to compare between the virulence repertoires of them. The six studied virulence genes were detected in UPEC isolates while only 4 genes were detected in DEC isolates. The most frequently detected virulence gene in UPEC and DEC isolates was *fimH* (UPEC isolates: n = 69/103, 66.9%; DEC isolates: n = 64/70, 91.4%). The frequencies of *iroN* gene were detected in a similar percentage for both types of isolates; (UPEC: 37/103, 35.9%; DEC: 25/70,

**Table 1. Primers sequences used for PCR assays.**

| Genes | Primer sequence | Size fragment (bp) | Reference |
|---|---|---|---|
| CVD432 | CTGGCGAAAGACTGTATCAT<br>CAATGTATAGAAATCCGCTGTT | 630 | [20] |
| ChuA | F-GACGAACCAACGGTCAGGAT<br>R-TGCCGCCAGTACCAAAGACA | 279 | [21] |
| yjaA | F-TGAAGTGTCAGGAGACGCTG<br>R-ATGGAGAATGCGTTCCTCAAC | 211 | [21] |
| TspE4C2 | F-GAGTAATGTCGGGGCATTCA<br>R-CGCGCCAACAAAGTATTACG | 154 | [21] |
| fimH | F: TGCAGAACGGATAAGCCGTGG<br>R: GCAGTCACCTGCCCTCCGGTA | 506 | [23] |
| Sfa (sfa/foc) | F: CTCCGGAGAACTGGGTGCATCTTAC<br>R: CGGAGGAGTAATTACAAACCTGGCA | 410 | [24] |
| pap A | F: ATGGCAGTGGTGTTTTGGTG<br>R:CGTCCCACCATACGTGCTCTTC | 720 | [23] |
| pap E/F | F: GCAACAGCAACGCTGGTTGCATCAT<br>R: AGAGAGAGCCACTCTTATACGGACA | 336 | [25] |
| hly A | F: AACAAGGATAAGCACTGTTCTGGCT<br>R: ACCATATAAGCGGTCATTCCCGTCA | 1170 | [25] |
| iroN | F AAGTCAAAGCAGGGGTTGCCCG<br>R GACGCCGACATTAAGACGCAG | 665 | [26] |

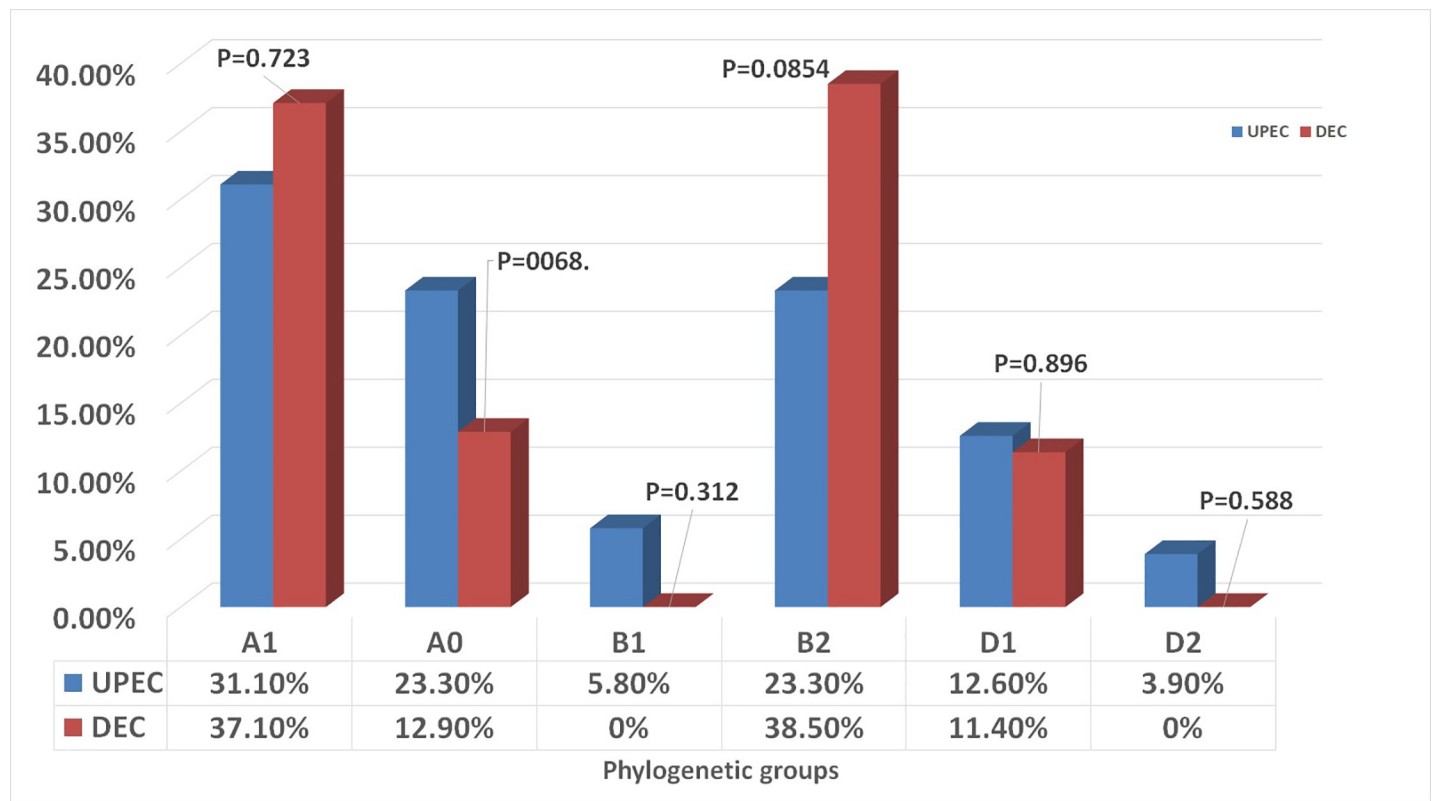

**Fig 1. Phylogenetic distribution of UPEC and DEC isolates.**

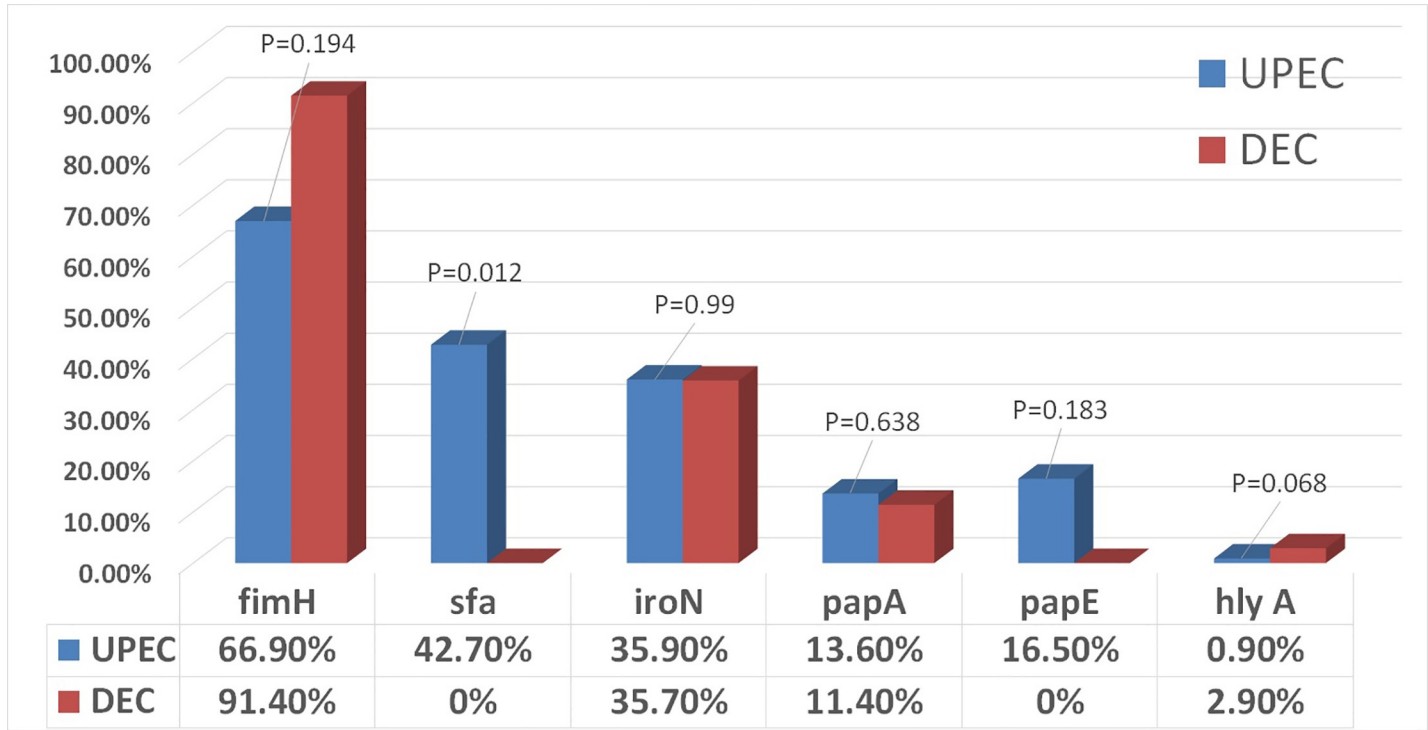

**Fig 2. Distribution of the virulence genes among UPEC & DEC isolates.**

35.7%). *papA* gene was detected in a percentage of (UPEC isolates: 14/103, 13.5%; DEC isolates: 8/70, 11.4%) and *hlyA* gene was found in only one isolate of UPEC (0.9%) and 2 isolates of DEC (2.9%). The *sfa* and *papE* genes were detected in UPEC isolates in (44/103, 42.7%) and (17/103, 17.1%) respectively but were not detected in DEC isolates at all. There were no significant differences between virulence genes in UPEC and DEC isolates, except for *sfa* gene (p value = 0.012) (Fig 2 and S2–S6 Figs).

Based on the distribution of the various genes, all the studied strains exhibited 21 virulence gene patterns, referred to as Ec (Table 2). UPEC isolates showed 20 Ec patterns while DEC isolates showed 5 Ec patterns only. Ec1 was characterized by the presence of the *fimH* gene only which was the most noted pattern and found in 24 UPEC isolates (23.3%) and 44 DEC isolates (62.9%). Ec3 (*fimH & iroN* genes) was the most frequent combined virulence pattern (11, 10.67% of UPEC and 14, 20%) of DEC isolates) followed by Ec5 (*fimH &sfa* genes; 9, 8.7% of UPEC isolates) and Ec10 (*fimH, papE &iroN*), (3, 2.91% of UPEC; 6, 8.57% of DEC isolates. The distribution of virulence patterns in phylogenic groups are summarized in Table 2.

## Discussion

Each *E.coli* type has characteristic phylogenetic and virulence patterns, which allow them to colonize and invade their host [14]. The phylogenetic analysis classify *E.coli* pathotypes into 4 major phylogenetic groups A, B1, B2 and D [13]. Several previous studies have reported that B2 and D phylogroups are the most common among UPEC isolates [27, 28], while isolates in phylogenetic groups A and B1 were mostly identified as commensal *E.coli* isolates [29]. Our study has reported different findings, where the predominant phylogenetic group in UPEC isolates was $A_1$ (31.1%), followed by $B_2$ (23.3%), $A_0$ (23.3%), and lastly $D_1$ (12.6%), that agree with some previous studies [30, 31]. The predominance of phylogenetic group A in UPEC

**Table 2. Virulence patterns (EC) distribution among Phylogenetic groups of UPEC and DEC *E.coli* isolates.**

| Virulence pattern | Phylogenetic group | | | | | | | Isolates N (%) | |
|---|---|---|---|---|---|---|---|---|---|
| | $A_0$ | $A_1$ | $B_1$ | $B_{2\,2}$ | $B_{2\,3}$ | $D_1$ | $D_2$ | UPEC | DEC |
| Ec1(UPEC/DEC) | 9 /5 | 8/16 | 1/- | 2/6 | 1/11 | 3/6 | - | 24 (23.3%) | 44 (62.9%) |
| Ec2 | 2 | 2 | - | 1 | 6 | 1 | - | 12(11.65%) | 0 |
| Ec3(UPEC/DEC) | 2/2 | 5/8 | 1 | -/2 | 1/2 | 2 | - | 11(10.67%) | 14 (20%) |
| Ec4(UPEC/DEC) | 5/2 | 3/2 | 1 | - | - | 1 | 1 | 11(10.67%) | 4 (5.71%) |
| Ec5 | 3 | 5 | - | - | - | 1 | - | 9 (8.7%) | 0 |
| Ec6 | - | 2 | - | 2 | 2 | - | - | 6 (5.8%) | 0 |
| Ec7 | - | 1 | 2 | 2 | - | - | - | 5 (4.85%) | 0 |
| Ec8 | - | 1 | - | - | - | 2 | - | 3 (2.91%) | 0 |
| Ec9 | - | - | 1 | - | - | 2 | - | 3 (2.91%) | 0 |
| Ec10(UPEC/DEC) | - | - | - | 1/2 | -/2 | -/2 | 2 | 3 (2.91%) | 6 (8.57%) |
| Ec11 | - | - | - | - | - | 1 | 1 | 2 (1.94%) | 0 |
| Ec12 | - | 1 | - | 1 | - | - | - | 2 (1.94%) | 0 |
| Ec13 | 2 | - | - | - | - | - | - | 2 (1.94%) | 0 |
| Ec14 | - | - | - | - | 2 | - | - | 2 (1.94%) | 0 |
| Ec15 | - | - | - | 2 | - | - | - | 2 (1.94%) | 0 |
| Ec16 | - | 2 | - | - | - | - | - | 2 (1.94%) | 0 |
| Ec17 | - | 1 | - | - | - | - | - | 1 (0.97%) | 0 |
| Ec18 | - | 1 | - | - | - | - | - | 1 (0.97%) | 0 |
| Ec19 | 1 | - | - | - | - | - | - | 1 (0.97%) | 0 |
| Ec20 | - | - | - | - | 1 | - | - | 11 (0.97%) | 0 |
| Ec21(UPEC/DEC) | - | - | - | -/2 | - | - | - | 0 | 2 (2.86%) |
| total(UPEC/DEC) | 24/9 | 32/26 | 6/- | 11/12 | 13/15 | 13/8 | 4/- | 103(100%) | 70(100%) |

UPEC, uropathogenic *E.coli*; DEC, Diarrheagenic *E.coli*; Ec1include (*fimH gene)*; Ec2 (*sfa gene)*; Ec3 (*fimH and iroN genes)*; Ec4 (no genes detected); Ec5 (*fimH and sfa)*; Ec6 (*fimH, sfa, papE and iroN genes)*; Ec7 ((*fimH, sfa and iroN genes)*; Ec8 (*fimH and papA)*; Ec9 (*papE)*; Ec10 (*fimH, papA and iroN genes)*; Ec11 (*sfa and iroN genes)*; Ec12 (*fimH, sfa, papE)*; Ec13 (*sfa, papA and iroN genes)*; Ec14 (*sfa, papA, papE and iroN genes)*; Ec15 (*fimH, sfa, papA, papE and iroN genes)*; Ec16 ((*fimH, papE and iroN genes)*, Ec17 (*papA)*; Ec18 (*papE)*; Ec19 ((*fimH, sfa, papA)*; Ec20 (*fimH, sfa, iroN and hly)*; Ec21 (*fimH, iroN and hly genes)*.

isolates which is usually associated with commensal strains suggesting that the gastrointestinal tract is the main source of strains that colonize the urinary tract [32, 33]. The distribution of virulence genes and phylogenetic types varies among different countries, for example, group A was the most prevalent in Russia [34], and also in China [35] in UPEC, so our findings can be explained by geographical variation.

Phylogenetic group A was also the predominant in DEC isolates of the current study, where the frequency of $A_1$ type was (37.1%) and $A_0$ was (12.9%). The predominance of phylogenetic group A in DEC isolates suggesting that the source of gastrointestinal tract infection can be endogenous even in immunocompetent persons. Although several previous reports showed that group B2 *E.coli* strains are rare in fecal samples of healthy persons [36, 37], group B2 was recorded in a high percent (38.5%) in our DEC isolates so our data reveals that acquiring the group B2 strain is important for developing of infection. On the other hand, previous studies, which investigated DEC isolates in Costa Rica [38] and Peru [39], showed that most of the isolates belonged to B1 and D groups respectively that disagrees with our study and reflects the diversity of DEC isolates in different countries. In order to find possible link between strain phylogeny and virulence genes, we analyzed an overall virulence profile (*fimH*, *sfa*, *iroN*, *pap A*, *papE* and *hly A* genes) in UPEC and DEC strains. The six studied virulence genes were

detected in UPEC isolates while only 4 genes were detected in DEC isolates. The most frequent gene was *fimH* gene (66.9% of UPEC isolates and 91.4% of DEC). A similar prevalence (65.9%) was detected in a previous study in Egypt [40] and also in other countries [41, 42]. For the other virulence genes *sfa*, *iroN*, *papA*, *papE* and *hly A*; the prevalence in our study was 42.7%, 35.9%, 13.6%, 16.5%, and 0.9% respectively in UPEC isolates. Our results were near to some studies in percent of *sfa* and *iroN* but the percent of *pap* and *hlyl* genes was lower [42]. Our results showed that the UPEC strains have different virulence profiles compared with other studies, suggesting that the virulence profile depends on the regional geography and climate or may be other factors.

The frequencies of urovirulence genes, *iroN* and *papA* detection in DEC were similar to that of UPEC isolates. There were no significant differences between virulence genes in UPEC and DEC isolates, except for *sfa* gene (p value = 0.012). About 66/70, 94.3% of DEC isolates carry at least one urovirulence gene. These findings showed a closer relationship between DEC specially EAEC and UPEC that may be explained by the remarkable genome plasticity or gene transfer in *E.coli* strains which leads to appearance of virulent strains displaying virulence genes from different pathotypes in a single isolate [18]. This interesting finding has shown that the classical classification of strains into pathotypes of *E.coli* is limited and inaccurate [19]. Our results showed that virulence genes were more prevalent in phylogroups A, so group A can be pathogenic and the other phylogoups as well. These findings also provide different insights into classification of *E.coli* pathotypes. A close relationship between the DEC and UPEC strains was reported, showing that DEC particularly the EAEC pathotype is an emerging enteropathogen which can cause intestinal and extra -intestinal infections, particularly in developing countries.

## Supporting information

**S1 Fig. PCR product for different phylogenetic types.**
(TIF)

**S2 Fig. PCR product of Fim A gene (506bp).**
(TIF)

**S3 Fig. PCR product of hly A gene (1170bpbp) and sfa gene (410bp).**
(TIF)

**S4 Fig. PCR product of pap E gene (336bp).**
(TIF)

**S5 Fig. PCR product of pap A gene (720bp).**
(TIF)

**S6 Fig. PCR product of iroN A gene (665bp).**
(TIF)

## Acknowledgments

We would like to thank the staff members of Minia University hospital for helping with the collection of samples

## Author Contributions

**Formal analysis:** Rasha M. Khairy, Soha S. Abdelrahim.

**Investigation:** Hend M. Abdel Ghany.

**Methodology:** Rasha M. Khairy, Ebtisam S. Mohamed, Soha S. Abdelrahim.

**Writing – original draft:** Rasha M. Khairy, Soha S. Abdelrahim.

**Writing – review & editing:** Rasha M. Khairy.

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
