## [Decision Letter · Decision Letter 0]

27 Jul 2019

PONE-D-19-16993

Phylogenic classification and virulence genes profiles of uropathogenic E. coli and diarrhegenic E. coli strains isolated from community acquired infections

PLOS ONE

Dear DR Khairy,

Thank you for submitting your manuscript to PLOS ONE. After careful consideration, we feel that it has merit but does not fully meet PLOS ONE’s publication criteria as it currently stands. Therefore, we invite you to submit a revised version of the manuscript that addresses the points raised during the review process.

In particular, a number of issues pertinent to the rationale of the study (Reviewer 1 - 1b) and the specifics on study groups utilized need to be included. This is in addition a number of other issues identified by both reviewers. 

We would appreciate receiving your revised manuscript by Sep 10 2019 11:59PM. To enhance the reproducibility of your results, we recommend that if applicable you deposit your laboratory protocols in protocols.io, where a protocol can be assigned its own identifier (DOI) such that it can be cited independently in the future. For instructions see: http://journals.plos.org/plosone/s/submission-guidelines#loc-laboratory-protocols

We look forward to receiving your revised manuscript.

Kind regards,

Praveen Thumbikat

Academic Editor

PLOS ONE

Journal Requirements:

1. Please provide additional details regarding participant consent. In the ethics statement in the Methods and online submission information, please ensure that you have specified (1) whether consent was informed and (2) what type you obtained (for instance, written or verbal). If your study included minors, state whether you obtained consent from parents or guardians. If the need for consent was waived by the ethics committee, please include this information.

Reviewers' comments:

Reviewer's Responses to Questions

**Comments to the Author**

1. Is the manuscript technically sound, and do the data support the conclusions?

Reviewer #1: Partly

Reviewer #2: Yes

2. Has the statistical analysis been performed appropriately and rigorously? 

Reviewer #1: Yes

Reviewer #2: Yes

3. Have the authors made all data underlying the findings in their manuscript fully available?

Reviewer #1: Yes

Reviewer #2: Yes

4. Is the manuscript presented in an intelligible fashion and written in standard English?

Reviewer #1: Yes

Reviewer #2: Yes

5. Review Comments to the Author

Reviewer #1: The paper entitled: “Phylogenic classification and virulence genes profiles of uropathogenic E. coli and

diarrhegenic E. coli strains isolated from community acquired infections” characterizes a number of E. coli isolates from diarrheal and urinary infections.

The main problems that I find in this paper are:

a) Virulence factors such as siderophores or adhesins, hemolysins, etc. are very common in commensal E. coli and it is not surprising that many DEC also have these genes. Having these genes doesn’t mean that the bacteria are UPEC. UPEC virulence is thought to be a multigenic phenomenon and describing all the potentially virulence genes is not very relevant.

b) It is not clear how they obtained the EAEC isolates; if they obtained from fecal cultures is very odd that all diarrheic samples were positive to EAEC. Normally you obtain 5-20% of different DEC pathotypes. If strains were obtained from a collection, why the authors concentrate in EAEC and ignore DEC pathotypes?

c) Study group is not described either for UTI patients nor for diarrhea patients.

d) Many reports have shown that some intestinal pathotypes (DEC) can also cause urinary infection DAEC and EAEC, it is not a novel finding.

e) The association of some DEC or UPEC strains in this study with Clermon’s phylogenetic groups is not uncommon. Virulence genes are transferred horizontally.

Minor problems:

a) The manuscript contains many grammatical errors.

b) The discussion section is too long

c) Figures showing distributions of strains in Clermon’s phylogenetic groups are unnecessary.

d) Lines 39-41 text improvement is needed

e) 107-126 Confidence intervals 95% are not reported for frequency data. There is not p value reported next to percentage of positiveness.

f) Table 2 and table 3 could be join in one.

g) 149-151 Use UPEC instead of ExPEC because you are working only with urine infections associated isolates.

h) 159 Country names must be written with uppercase

i) 157-161 Improve English writing

j) The format of bibliographical references should be checked

Reviewer #2: Interesting article. It brings information that can be taken into account in future studies and analysis at a global level. Evaluations within the framework of general data are, however, rather geared towards parts of the globe and less in Europe (eg could be seen doi.org/10.1155/2019/5712371). Please review the English language. I've seen some errors to mention just one (page 4, 77, enteroaggragative). Please review the bibliography - it is not unitary written.

6. PLOS authors have the option to publish the peer review history of their article (what does this mean?). If published, this will include your full peer review and any attached files.

Reviewer #1: No

Reviewer #2: No

---

## [Author Response · Author response to Decision Letter 0]

2 Aug 2019

Reviewer comments 1 : 

reviewer1` 

a) Virulence factors such as siderophores or adhesins, hemolysins, etc. are very common in commensal E. coli and it is not surprising that many DEC also have these genes. Having these genes doesn’t mean that the bacteria are UPEC. UPEC virulence is thought to be a multigenic phenomenon and describing all the potentially virulence genes is not very relevant.

Response

Bacteria are classified as intestinal pathogenic E. coli (IPEC), which are associated with diarrhea, and extraintestinal pathogenic E. coli (ExPEC), which cause infections beyond the intestinal tract) not only on the basis of their virulence properties but also the site of infection (the diseases that they cause),, and their host of isolation (Russo TA, Johnson JR. 2000; Kaper, J. B. 2005; Kaper, J. B. et al 2004). The study was to compare the virulence genes in both UPEC isolated from patients with UTI and DEC isolated from patients suffering from diarrhea in Egypt, where the data is scarce. Our isolates were recovered from diagnosed clinical infections and from pure cultures, so they associated with the diseases caused by them

 The hypothesis express, that uropathogenic E. coli (UPEC) strains than non-pathogenic strains acquiring new virulence factors by pathogenicity islands (Mladin et al., 2009).

Many reports lie in the same corner with us; (Connell I, et al. 1996) who reported that Expression of type 1 fimbriae (adhesin gene fimH ) has been closely associated with the early development of UTI (Connell I, et al. 1996. Proc. Natl. Acad. Sci. U. S. A. 93:9827–9832.) (Johnson JR and Russo TA. 2002; Köhler CD and Dobrindt U. 2011. And others) have demonstrated that the main ExPEC virulence factors, such as P fimbriae and α-hemolysin, are usually not present in intestinal pathogenic (IPEC) isolates.

Lee et al. (2010) found that among commensal strains, virulence factors were good predictors for urinary tract or blood stream infections, while Peirano et al. (2013) defined isolates positive for two or more of papA and/or papC, sfa/focDE, afa/draBC, kpsM II and iutA as ExPEC.

Our results agree with that UPEC virulence is thought to be a multigenic phenomenon as we found UPEC isolates exhibited 20 Ec patterns while DEC isolates exhibited 5 Ec patterns only. Combined virulence patterns are in UPEC. Some strains of E. coli can diverge from their commensal cohorts, taking on a more pathogenic nature. These strains acquire specific virulence factors (horizontal transfer of transposons, plasmids, bacteriophages, and pathogenicity islands).

- 

- We also found that some virulence genes are not common in DEC isolates. sfa and papE genes were detected in UPEC isolates in (44/103, 42.7%) and (17/103, 17.1%) respectively but were not detected in DEC isolates at all. 

VGs are ideal targets for determining the pathogenic potential of a given E. coli isolate (Kuhnert P, Boerlin P, Frey J. 2000. FEMS Microbiol. Rev. 24:107–117)

b) It is not clear how they obtained the EAEC isolates; if they obtained from fecal cultures is very odd that all diarrheic samples were positive to EAEC. Normally you obtain 5-20% of different DEC pathotypes. If strains were obtained from a collection, why the authors concentrate in EAEC and ignore DEC pathotypes?

We concentrated in EAEC because it is the most common type in Egypt (Ali et al., 2014; El Gamel et al., 2015). Other types are less frequent and their number in most of collections is not representative. 

c) Study group is not described either for UTI patients or for diarrhea patients.

We thank the reviewer for comment. The revised manuscript included the reviewer's suggestion.

d) Many reports have shown that some intestinal pathotypes (DEC) can also cause urinary infection DAEC and EAEC, it is not a novel finding.

There are no studies done in Egypt to associate between DEC isolates and UPEC isolates. So we tried to study this issue in a new geographical area. Only one or 2 reports have studied the virulence factors of DEC (eae, bfp) in UTI.

The pathogenic E. coli strains use a complex multistep mechanism of pathogenesis involving a number of virulence factors depending upon the pathotype, which consists of attachment, host cell surface modification, invasin, a variety of toxins, and secretion systems which eventually lead toxins to the target host cells (kapper. 2004). Thus, VGs are ideal targets for determining the pathogenic potential of a given E. coli isolate (Kuhnert P, Boerlin P, Frey J. 2000).

e) The association of some DEC or UPEC strains in this study with Clermon’s phylogenetic groups is not uncommon. Virulence genes are transferred horizontally.

The phylogenetic grouping was done to identify the prevalence of different phylogenetic groups in Egypt and comparing between UPEC and DEC isolates to increase the knowledge about the geographical distributions of these types. In contrast to most reports, phylogenetic group A was the most prevalent in both UPEC and DEC strains, followed by B2 group. The predominance of phylogenetic group A which is usually associated with commensal strains suggesting that the gastrointestinal tract is the main source of strains that may be colonize the urinary tract. This increases and highlights the emergence of virulent phylogroups A and B1. This result was similar to some previous studies (Rahman et al., 2017). Although other studies showed the more prevalence of phylogenetic groups B2 and D (ElSayed Gawad et al., 2018). It was surprising that B1 type was not detected in DEC isolates at all.

Minor problems

a) The manuscript contains many grammatical errors.

We thank the reviewer for comment. The revised manuscript included the reviewer's suggestion

b) The discussion section is too long

We thank the reviewer for comment. The revised manuscript included the reviewer's suggestion

c) Figures showing distributions of strains in Clermon’s phylogenetic groups are unnecessary.

We thank the reviewer for comment. The revised manuscript included the reviewer's suggestion

d) We thank the reviewer for comment. The revised manuscript included the reviewer's suggestion

 Lines 39-41 text improvement is needed

Done

e) 107-126 Confidence intervals 95% are not reported for frequency data. There is not p value reported next to percentage of positiveness.

We thank the reviewer for comment. The revised manuscript included the reviewer's suggestion

f) Table 2 and table 3 could be join in one.

We thank the reviewer for comment. The revised manuscript included the reviewer's suggestion

g) 149-151 Use UPEC instead of ExPEC because you are working only with urine infections associated isolates.

Ok 

h) 159 Country names must be written with uppercase

Ok 

i) 157-161 Improve English writing

We thank the reviewer for comment. The revised manuscript included the reviewer's suggestion

j) The format of bibliographical references should be checked

We thank the reviewer for comment. The revised manuscript included the reviewer's suggestion

Reviewer #2: Interesting article. It brings information that can be taken into account in future studies and analysis at a global level. Evaluations within the framework of general data are, however, rather geared towards parts of the globe and less in Europe (eg could be seen doi.org/10.1155/2019/5712371). Please review the English language. I've seen some errors to mention just one (page 4, 77, enteroaggragative). Please review the bibliography - it is not unitary written.

 We thank the reviewer for comment. The revised manuscript included the reviewer's suggestion

---

## [Decision Letter · Decision Letter 1]

30 Aug 2019

[EXSCINDED]

Phylogenic classification and virulence genes profiles of uropathogenic E. coli and diarrhegenic E. coli strains isolated from community acquired infections

PONE-D-19-16993R1

Dear Dr. Khairy,

We are pleased to inform you that your manuscript has been judged scientifically suitable for publication and will be formally accepted for publication once it complies with all outstanding technical requirements.

With kind regards,

Praveen Thumbikat

Section Editor

PLOS ONE

Additional Editor Comments (optional):

Reviewers' comments:

Reviewer's Responses to Questions

**Comments to the Author**

1. If the authors have adequately addressed your comments raised in a previous round of review and you feel that this manuscript is now acceptable for publication, you may indicate that here to bypass the “Comments to the Author” section, enter your conflict of interest statement in the “Confidential to Editor” section, and submit your "Accept" recommendation.

Reviewer #2: All comments have been addressed

2. Is the manuscript technically sound, and do the data support the conclusions?

Reviewer #2: Yes

3. Has the statistical analysis been performed appropriately and rigorously? 

Reviewer #2: Yes

4. Have the authors made all data underlying the findings in their manuscript fully available?

Reviewer #2: Yes

5. Is the manuscript presented in an intelligible fashion and written in standard English?

Reviewer #2: Yes

6. Review Comments to the Author

Reviewer #2: (No Response)

7. PLOS authors have the option to publish the peer review history of their article (what does this mean?). If published, this will include your full peer review and any attached files.

Reviewer #2: No

---

## [Editor Report · Acceptance letter]

4 Sep 2019

PONE-D-19-16993R1 

Phylogenic classification and virulence genes profiles of uropathogenic *E. coli* and diarrhegenic *E. coli*  strains isolated from community acquired infections 

Dear Dr. Khairy:

I am pleased to inform you that your manuscript has been deemed suitable for publication in PLOS ONE. Congratulations! Your manuscript is now with our production department. 

With kind regards,

on behalf of

Dr. Praveen Thumbikat 

Section Editor

PLOS ONE